# Personality Traits Predict 7-Year Risk of Diagnosis of Multiple Sclerosis: A Prospective Study

**DOI:** 10.3390/jcm12020682

**Published:** 2023-01-15

**Authors:** Weixi Kang

**Affiliations:** UK DRI Care Research and Technology Centre, Department of Brain Sciences, Imperial College London, 926, Sir Michael Uren Hub, 86 Wood Lane, London W12 0BZ, UK; weixi20kang@gmail.com; Tel.: +44-73-980-44185

**Keywords:** multiple sclerosis, personality, Big Five, Openness, Conscientiousness

## Abstract

Objective: The objective of the current study is to investigate how Big Five personality traits could predict the risk of multiple sclerosis (MS) diagnosis in 7 years. Methods: A binary logistic regression was used to analyze data from 17,791 participants who responded to questions at Wave 3 (collected between 2011 to 2012) and Wave 10 (collected between 2018 to 2019) using a binary logistic regression from UKHLS with a mean age of 47.01 (S.D. = 16.31) years old with 42.62% males. Results: The current study found that Openness (OR = 0.68, *p* < 0.01, 95% C.I. (0.51, 0.89)) and Conscientiousness (OR = 0.70, *p* < 0.05, 95% C.I. (0.52, 0.93)) are positively associated with a reduced risk of MS diagnosis in 7 years. Conclusion: Health professionals can use findings from the current study as evidence for developing tools for assessing the risk of MS, and providing interventions for people who may be at high risk of MS based on their personality traits.

## 1. Introduction

Personality refers to a set of enduring traits that can be reflected in one’s feelings, thoughts, and behaviors, which have been proven to be stable over time. The Big Five model of personality consists of five broad traits/domains including Neuroticism, Agreeableness, Openness, Conscientiousness, and Extraversion [1,2]. Most psychologists agree that the Big Five model of personality can capture the most basic level of individual differences and other personality models can be translated into the Big Five personality model [3,4,5]. Specifically, Neuroticism refers to the tendency of being emotionally unstable and having higher chances of experiencing negative affect, depression, anxiety, and psychological distress. Agreeable people tend to be cooperative, altruistic, and empathic. People with high Openness tend to have various interests, appreciate arts and beauties, and prefer novelty to routine. Conscientiousness refers to the tendency of being task-focused, organized, and self-disciplined. Finally, Extraversion refers to the tendency of being talkative, assertive, and sociable.

Personality traits are agreed to be predictors of health [6]. The relationships between personality traits and health can even be held across decades, as shown by the finding that personality traits in childhood are predictive of self-rated health in middle age [7]. Moreover, these findings can extend beyond self-ratings of health [8] to more objectively assessed health measures, such as physician-rated health [9], biomarkers of health [10], and longevity [11,12,13]. There has been growing interest in exploring the predictive value of personality traits in various diseases. For instance, there are prospective associations between Big Five personality traits and disease later, such as the risk of stroke, lung disease, and heart condition [14].

Multiple sclerosis (MS) is a chronic neurodegenerative disease of the central nervous system. The prevalence of MS has recently increased, affecting around 2.8 million people in the world [15]. There have been some cross-sectional studies that documented the differences in personality traits between patients with MS and people without MS [16,17,18,19,20,21]. However, few of them used the Big Five model to measure personality traits [22]. The most typical finding is that patients with MS tend to have increased Neuroticism [19,20,21]. One study has also suggested that MS patients have lower Conscientiousness [16,23]. found higher Neuroticism but lower Agreeableness and Extraversion in a small sample of MS patients. However, some studies at a large scale found minimal or/or non-significant differences in personality traits between patients with MS and healthy controls [24,25,26].

Thus, although there are some cross-sectional findings that compare differences in personality traits between MS patients and healthy controls, these associations can be bidirectional as personality traits may be risk factors for MS, and MS can lead to brain damage that causes personality change. It remains largely unclear regarding how Big Five personality traits could prospectively predict the risk of MS. The aim of the current study is to explore how the Big Five personality traits could predict the diagnosis of MS in 7 years.

## 2. Methods 

### 2.1. Data

This study used data from Understanding Society: the UK Household Longitudinal Study (UKHLS), which has been collecting annual information from the original sample of UK households since 1991 [27]. Participants completed the demographics and personality questions at Wave 3, which was collected between 2011 and 2012, and questions regarding if they have been clinically diagnosed with MS at Wave 10, which was collected between 2018 and 2019. All data collections have been approved by the University of Essex Ethical Committee. Participants completed informed consent agreements before participation in all studies. This data set is publicly available at https://www.understandingsociety.ac.uk (accessed on 1 September 2022). Participants with any missing variables of interest were removed from further analysis. Thus, there were 17,791 participants with a mean age of 47.01 (S.D. = 16.31) years old with 42.62% males remaining for further analysis.

### 2.2. Measures

#### 2.2.1. Personality Traits

Personality was measured using the 15-item version of the Big Five Inventory, with a Likert scale ranging from 1 (“disagree strongly”) to 5 (“agree strongly”). Scores were reverse coded when appropriate. The exact set of questions used to ask participants can be found at https://www.understandingsociety.ac.uk/documentation/mainstage/dataset-documentation/term/personality-traits?search_api_views_fulltext= (accessed on 1 September 2022). Mean scores were used for each of these traits. All personality scores were standardized (mean = 0, SD = 1) before further analysis. 

#### 2.2.2. MS

Participants answered the question “Has a doctor or other health professional ever told you that you have any of these conditions? Multiple sclerosis.” to indicate if they have been diagnosed with MS. Self-reported MS is a valid measure of MS status and has been used in various studies [28,29].

#### 2.2.3. Demographic Controls

Demographic controls in the model include age (continuous), sex (male = 1 vs. female = 2), monthly income (continuous), highest educational qualification (college = 1 or below college = 2), legal marital status (single = 1 vs. married = 2), and residence (urban = 1 vs. rural = 2).

### 2.3. Analysis

Personality traits including Neuroticism, Agreeableness, Openness, Conscientiousness, and Extraversion scores were standardized before (mean = 0, S.D. = 1) further analysis. A binary logistic regression was used by taking demographics including age, sex, monthly income, highest educational qualification, legal marital status, residence, and standardized personality traits scores at Wave 3 predictors to predict ever diagnosis of MS at Wave 10.

## 3. Results

Descriptive statistics can be found in Table 1. The current study found that being a female (OR = 2.37, *p* < 0.01, 95% C.I. (1.29, 4.36)) and receiving at least college education (OR = 2.42, *p* < 0.01, 95% C.I. (1.41, 4.15)) are positively related to the risk of MS diagnosis in 7 years, whereas Openness (OR = 0.68, *p* < 0.01, 95% C.I. (0.51, 0.89)) and Conscientiousness (OR = 0.70, *p* < 0.05, 95% C.I. (0.52, 0.93)) are associated with a reduced risk of MS diagnosis in 7 years (Table 2).

## 4. Discussion

Taken together, the aim of this prospective study was to test whether Big Five personality traits are predictive of MS diagnosis in 7 years. By analyzing data from 17,791 participants using a binary logistic regression from UKHLS, the current study found that Openness and Conscientiousness are prospectively associated with a reduced risk of MS diagnosis in 7 years.

The current study adds a substantial amount of evidence to the literature regarding the health benefits of Openness and Conscientiousness [30,31,32]. Indeed, recent studies have indicated that Openness is related to more physical activities [33], better physical functions [34,35], and low inflammations [36], which can then reduce the risks of MS diagnosis in 7 years [37].

The finding that higher Conscientiousness is associated with a reduced risk of MS diagnosis in 7 years agrees with a previous cross-sectional study regarding lower Conscientiousness scores among MS patients [16]. Conscientiousness is also positively related to health-promoting behaviors such as more physical activities [38], and fewer behaviors that may be detrimental to health, such as smoking and alcohol use [39,40,41]. In addition, Conscientiousness is associated with a lower risk of physiological dysfunction [42], inflammation [36], and depressive symptoms [43] over time, which are risk factors for MS [37]. This link can also be partially explained by biological factors. Indeed, Conscientiousness is associated with healthier metabolic, cardiovascular, and inflammatory markers [36,42], which may lead to a reduced risk of MS diagnosis in 7 years [37].

Despite the strengths of the current study including a prospective approach, and well-controlled sociodemographic characteristics, there are also some limitations. First, the current study did not control for well-identified risk factors of MS, including high latitude, smoking, low vitamin D levels, and Epstein–Barr virus (EBV) infection, which makes it hard to test whether these risk factors serve as potential mediators for the associations between personality traits and risk of MS diagnosis in 7 years [37]. Future studies should test the potential mediators that exist in the association between personality traits and the risk of MS diagnosis. Second, the current study focused on self-reported data. Future studies should use more objective assessments to measure personality traits and MS diagnosis. Third, the current study focused on samples from the United Kingdom, which may make it hard to generalize the current findings to other countries and contexts. Future studies should test if these associations still exist in other countries. Finally, the p-value of Conscientiousness was only below 0.5, so perhaps more data would be needed to confirm this finding.

In conclusion, this current study found that high Openness and Conscientiousness are related to a lower risk of MS diagnosis in 7 years, which supports the emerging evidence regarding the role of personality traits in the health process [6,44] and as risk factors for chronic diseases. Health professionals can use findings from the current study as evidence for developing tools for assessing the risk of MS and providing interventions for people who may be at high risk of MS based on their personality traits.

## Figures and Tables

**Table 1 jcm-12-00682-t001:** Descriptive statistics for demographics and personality traits.

	Mean	S.D.
Age	47.01	16.31
Monthly income	1465.90	1381.38
Neuroticism	3.58	1.43
Agreeableness	5.64	1.02
Openness	4.58	1.28
Conscientiousness	5.51	1.08
Extraversion	4.58	1.30
	**N**	**%**
**Sex**		
Male	7583	42.62
Female	10,208	57.38
**Highest educational qualification**		
Below college	11,752	66.06
College	6039	33.94
**Legal marital status**		
Single	7848	44.11
Married	9943	55.89
**Residence**		
Urban	13,423	74.45
Rural	4368	24.55
**MS at Wave 10**		
Yes	17,727	99.64
No	64	0.36

**Table 2 jcm-12-00682-t002:** The odd ratios of the probability of demographic and personality traits predictors in predicting the chance of MS diagnosis in 7 years.

	OR	P	95% C.I.
Age	1.00	0.72	(0.99, 1.02)
Sex	2.37	<0.01	(1.29, 4.36)
Monthly income	1.00	0.59	(1.00, 1.00)
Highest educational qualification	2.42	<0.01	(1.41, 4.15)
Marital status	0.74	0.25	(0.45, 1.23)
Residence	1.11	0.72	(0.63, 1.97)
Neuroticism	0.88	0.40	(0.67, 1.17)
Agreeableness	1.09	0.56	(0.81, 1.47)
**Openness**	**0.68**	**<0.01**	**(0.51, 0.89)**
**Conscientiousness**	**0.70**	**<0.05**	**(0.52, 0.93)**
Extraversion	1.09	0.55	(0.82, 1.44)

## Data Availability

Publicly available datasets were analyzed in this study. This data can be found here: https://www.understandingsociety.ac.uk.

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
