# Peer review of "Personality Traits Predict 7-Year Risk of Diagnosis of Multiple Sclerosis: A Prospective Study"

_jcm, 2023, doi:10.3390/jcm12020682_

Round 1

Reviewer 1 Report

Authors have tried to find the role of personality traits in predicting risk of MS over 7 years.

Interesting study from a neurologist point of view.

Personality traits can be considered as an association but not causation. More data is needed to prove the causation. Hence, one cannot infer that having specific personality would led to MS. Moreover, demyelinating plaques can themselves lead on to cognitive and personality changes. A simple way to find the link if we compare the personality traits of patients with discrete plaques corresponding to deficit versus patients presenting with high burden of plaques.

Draft is well written and appropriate for brief report section.

Please give appropriate reference: “Özmen & Yurtta., 2018” is about caregiver burden, not incidence or prevalence of MS.

Regards

Author Response

Thanks for reviewing my manuscript and for your comments. I can confirm that I have now cited an appropriate reference for multiple sclerosis prevalence. 

Reviewer 2 Report

A very interesting paper which is a good contribution to the landscape. 

The fact that personality traits may be associated with better outcomes in MS is very interesting and adds to the need for an holistic approach to care. 

My only comment was around the number variables (11) which were processed against diagnosis and the statistical significance being set at 0.05. It could be possible for someone who was being harsh on the paper to accuse it of verging on data dredging. An acknowledgement in the limitations of this would settle this issue. 

Overall though, an interesting and well written paper. Well done!

Author Response

Thanks for reviewing my manuscript and for your comments. I have now added your suggestions as a limitation.